

# In situ observations of meteorological variables and snowpack distribution at the Izas Experimental Catchment (Spanish Pyrenees): The importance of high quality data in sub-alpine ambients.

Jesús Revuelto[1,2], Cesar Azorin-Molina[1,3], Esteban Alonso-González[1], Alba Sanmiguel-
Vallelado[1], Francisco Navarro-Serrano[1], Ibai Rico[1,4], Juan Ignacio López-Moreno[1]

1 Pyrenean Institute of Ecology, CSIC, Zaragoza, Spain
2 Météo-France - CNRS, CNRM (UMR3589), Centre d'Etudes de la Neige, Grenoble, France
3 Regional Climate Group, Department of Earth Sciences, University of Gothenburg, Gothenburg, Sweden
4 University of the Basque Country. Department of Geography, Prehistory and Archaeology. Vitoria, Spain
Correspondence to: Jesús Revuelto jesus.revuelto@meteo.fr

**Abstract:** This work describes the snow and meteorological dataset available for the Izas Experimental Catchment, in the Central Spanish Pyrenees, from 2011 to 2016 snow seasons. The experimental site is located in the southern side of the Pyrenees between 2000 and 2300 m above sea level with an extension of 55 ha. The site is a good example of sub-alpine ambient in which snow accumulation and melting dynamics have major importance in many mountain processes. The climatic dataset includes information on different meteorological variables acquired with an Automatic Weather Station (AWS) such as precipitation, air temperature, incoming and reflected short and long-wave radiation, relative humidity, wind speed and direction, atmospheric air pressure, surface temperature (snow or soil surface) and soil temperature; all of them at 10 minute intervals. Snow depth distribution was measured during 23 field campaigns using a Terrestrial Laser Scanner (TLS), and there is also available daily information of the Snow Covered Area (SCA) retrieved from time-lapse photography. The data set (https://doi.org/10.5281/zenodo.579979) is valuable since it provides high spatial resolution information on the snow depth and snow cover distribution, which is particularly useful in combination with meteorological variables to simulate the snow energy and mass balance. This information has already been analyzed in different scientific works studying snow pack dynamics and its interaction with the local climatology or terrain topographic characteristics. However, the database generated till the date has great potential for understanding other environmental processes from a hydrometerological or ecological perspective in which snow dynamics play a determinant role.



## 1. Introduction

The snowpack distribution and its temporal evolution have a marked influence in many mountain processes. For instance, erosion rates and sediment transport (Colbeck et al., 1979; Lana-Renault et al., 2011), geomorphological and glaciological processes (López-Moreno et al., 2015; Serrano et al., 2001), phenological cycles (Liston, 1999; Wipf et al., 2009) are directly controlled by the timing of snow distribution. In the other hand, snow melting dynamics also has major importance from a hydrological perspective since one-sixth of total Earth's population depends on the water storage in mountain rivers headwaters (Barnett et al., 2005). In downstream areas exposed to extreme climatic conditions, the snowmelt runoff from mountain areas becomes a key element (Viviroli et al., 2007), especially in these zones subjected to water scarcity. Such is the case of semi-arid regions, as the Mediterranean area which is characterized by an irregular climate with long drought periods (Vicente-Serrano, 2006), and thereby its dependence on water stored in mountain areas as the Pyrenees is quite high (López-Moreno, 2005; López-Moreno et al., 2008).

The Pyrenees are a mid-latitude mountain range, with significant snow presence in high elevation areas along the year. During the boreal spring, Pyrenean river discharges depend on the melting of snow accumulated in previous months, directly accounting from snow about 40% of spring runoff (López-Moreno and García-Ruiz, 2004). Thus, snow accumulation has a large influence on Pyrenean headwaters. This dependence is rather due to the generally continuous snow cover from November to April above 2000 m above sea level (a.s.l.) (Alvera and Garcia-Ruiz, 2000; García-Ruiz et al., 1986; López-Moreno et al., 2001). This way the study of the snowpack in high elevation areas of the Pyrenees is crucial for understanding and managing mountain river discharges (López-Moreno, 2005), especially in the frame of a global change scenario (García-Ruiz et al., 2011). However, the existence of continuous snow observations above 2000 m a.s.l. is scarce in this mountain range, being most of them available from 1600 to 2000 m a.s.l. and when available these observations spam over short time periods. Thereby well-established study areas in high elevation, having continuous measurements of meteorological variables and snowpack distribution are required in the Pyrenees.

In this paper, it is presented the recently acquired dataset of meteorological and snowpack variables obtained in a small size experimental catchment of the southern side of the Pyrenees. Although meteorological and hydrological data are available since



previous years (some variables were measured since late 80's (Alvera and Garcia-Ruiz, 2000)), we offer data from 2011/12 to 2015/16 snow season, as data series have higher quality and continuity, and also they match with in situ observations of snow depth and snow cover. The dataset consist of (*i*) continuous meteorological variables acquired from an Automatic Weather Station (AWS), (*ii*) detailed information on snow depth distribution collected with a Terrestrial Laser Scanner (TLS, LiDAR technology) for certain dates along the snow season (between 3 and 6 TLS surveys per snow season) and (*iii*) time-lapse images that show the Snow Covered Area evolution (SCA).

The paper is structured as follows: Section 2 describes the study area characteristics; Section 3 presents meteorological data acquired from the AWS with a general description of the observed climatology; Section 4 describes the distributed measurements on snow depth distribution from the TLS and the SCA derived from time-lapse images; Section 5 concludes with information for downloading the database; and finally Section 6 summarizes all information available and the potential application of the database

## 2. Study area characteristics and climatology

### 2.1. The Pyrenees

The Pyrenees lies in the northeastern limit of the Iberian Peninsula (Figure 1). It is an orographic barrier between north and south face. This way a progressively higher aridity is found southward as a consequence of the mountain range blocking humid air masses from the Atlantic (López-Moreno and Vicente-Serrano, 2007; Vicente-Serrano, 2005). Thus, the natural barrier directly influence precipitation and as a consequence areas above 2000 m a.s.l. receive about 2000 mm/year, increasing to 2500 mm/year in the highest divides of the mountain range and rapidly decrease to 600-800 mm/year in low elevation areas of the southern side (García-Ruiz, et al., 2001).

Another distinct feature of the Pyrenees is their location between two water masses with contrasted conditions; i.e., in the western side is the Atlantic Ocean, while in the east side lays the Mediterranean Sea. This situation between both water masses originates a climatic transition from Oceanic to Mediterranean conditions to the east. During autumn, fronts approaching from the Atlantic bring the highest monthly averages of precipitation in the western observatories, reaching a 40% of total annual precipitation in this area (Creus-Novau, 1983). Oppositely, spring and summer storms mostly affect



the eastern areas of the Pyrenees, being favored by the development of sea breeze and local wind convergence zones that initiate deep moist convection along the eastern fringe of the Iberian Mediterranean area (Azorin-Molina et al., 2015). Thereby Pyrenean observatories in the east have a major contribution of convective events; e.g., reaching a

32% of total annual precipitation in eastern valleys and dropping below 16% of annual precipitation in western valleys (Cuadrat et al., 2007). In early winter, the arrival of fronts from northwest and west are the most frequent, leading to highest snow accumulation in the western Pyrenees (Navarro-Serrano and López-Moreno, 2017). The Azores high, which usually affects the Iberian Peninsula for some winter periods,

originates relatively long periods with no snow accumulation in this season. Subsequently, in spring, snow accumulation are associated with southwesterly advections, which lead to high snow accumulations in the western Pyrenees (Revuelto et al., 2012). Snow remains for long periods above 1600 m a.s.l., between November and April (López-Moreno and Nogués-Bravo, 2006).

Similarly to precipitation, air temperature is influenced by the Atlantic-Mediterranean transitions, but elevation plays a major role on its distribution. For instance, the lower annual thermal amplitude is observed in the western Pyrenees because ocean proximity (Cuadrat et al., 2007). As a general tendency in the Central Pyrenees, the annual 0ºC isotherms lays between 2700 and 2900 m a.s.l. (del Barrio et al., 1990; Chueca, J.,

1993).

Additionally the Pyrenees exhibit a high inter-annual variability in air temperature and precipitation, which involve great uncertainty in annual snow accumulation (López-Moreno, 2005). This variability is influenced by the inter-annual variability of atmospheric circulation, being identified a decrease of snow accumulation weather

types under positive North Atlantic Oscillation (NAO) phases (López-Moreno and Vicente-Serrano, 2007). As observed with precipitation, snow accumulation correlates with Atlantic-Mediterranean proximity and distance to the main divide of the mountain range (Revuelto et al., 2012), and is strongly dependent on the fluctuations of the 0ºC isotherm during winter and spring. This high climatic variability also originates a large

inter-annual variability in total snow accumulation and on its temporal distribution along the snow season (López-Moreno, 2005).

### 2.2. The Izas Experimental Catchment

The Izas Experimental Catchment (42°44′ N, 0°25′ W) has a surface of 33 ha, but snow depth information cover a total of 55 ha, with elevations ranging between 2075 and





2325 m a.s.l. This area is close to the main divide of the Pyrenees in the headwaters of the Gállego River, near the Spain–France border (Figure 1). The Izas Experimental Catchment exemplifies the general characteristics of sub-alpine areas of the Pyrenees. In this environment, snowpack dynamics have a major importance along the year. Thus the atmosphere-snowpack interactions observed at this experimental site will enable to better understand many processes of sub-alpine areas.

The mean annual precipitation is 2000 mm, and snow accounts for approximately 50% of total precipitation (Anderton et al., 2004). For an average of 130 days each year the mean daily air temperature is below 0 ºC, with a mean annual air temperature of 3 ºC, (del Barrio et al., 1997). Snow covers a high percentage of the catchment from November to the end of May (López-Moreno et al., 2010). Lithology shows limestones and sandstones of the Cretaceous period, and limestones of the Paleocene, much more resistant to erosion. Zonal vegetation corresponds to a high mountain steppe, mainly covered by bunch grasses, namely *Festuca eskia*, *Nardus stricta, Trifolium alpinum, Plantago alpine* and *Carex sempervirens.* Rocky outcrops dominate in the upper and steeper slopes (less than 15% of the study area). There are not trees present in the study area. The catchment is predominantly east-facing, with some areas also facing north or south. The mean slope of the catchment is 16° (López-Moreno et al., 2012), having the typical high spatial heterogeneity on its topographic characteristics of sub-alpine areas, having flat concave and convex areas.

## 3. Meteorological data

The study site is equipped with an AWS located in the lower elevation of the catchment (42º 44' 33.65''N, 0º 25' 8.83'', 2113 m a.s.l., Figure 1). The AWS measures wind speed and direction, atmospheric air temperature, relative humidity and air pressure, soil temperature for 0 cm, 5 cm, 10 cm and 20 cm, temperature of the surface close to the AWS (snow or soil, depending if snow is present or not), global and reflected solar irradiance, snow depth and precipitation (the precipitation gauge is located at 15 m of the AWS tower) (see Figure 2). Information on the main atmospheric variables has been recorded since the end of 2011 (AWS installed on November 2011). Therefore, data availability covers five complete snow seasons. Since the station is located in the lower elevation of the catchment and despite air temperature lapse rate with elevation, the





AWS records serve to average the evolution of atmospheric variables occurring at the Izas Experimental Catchment.

The data acquisition system consists of a Campbell Scientific CR3000 datalogger that samples each instrument and stores data at 10-minute time intervals. All data is

5 transmitted via modem to the Pyrenean Institute of Ecology and once received we apply some automatic quality-control checks for removing outliers. Data gaps are rare for almost all variables and therefore instead of gap-filling with interpolation methods, only measured data are available. However, some variables had long data gaps and thereby some periods have been discarded for further analysis. This is the case of precipitation

for the first three snow seasons which were useless because the length of data gaps.

Since the main application of the data collected by the AWS is the assessment of the snow cover evolution in the study area, in the following subsections we focus our analyses on the accumulation and melting periods: i.e., accumulation (January, February and March; JFM) and melting (April, May and June; AMJ) periods Annual values

observed during a whole snow season are also presented for each sub-section.

**3.1. Wind speed and direction**

The AWS is equipped with a Young wind monitor – ALPINE MODEL (Young Company, Model 05103-45-5; http://www.youngusa.com/Brochures/05103-45%20(0613).pdf ), placed in the highest point of the meteorological tower (8 m above

20 the ground). The Pyrenees are commonly affected by strong westerly to northerly winds as shown in the wind roses displayed in Figure 3. With the exception of south winds mainly occurring during the melting period, westerly to northerly winds dominate. Additionally, the frequency of moderate to strong winds mainly occurs for northwesterly winds.

**3.2. Air temperature, relative humidity and atmospheric air pressure**

Air temperature and relative humidity are measured with the HMP155 Vaisala sensor (http://www.vaisala.com/Vaisala%20Documents/Brochures%20and%20Datasheets/HMP155-Datasheet-B210752EN-E-LoRes.pdf), and atmospheric air pressure is recorded with the BP1 sensor from Adcon telemetry (http://www.adcon.com/download/leaflet-

30 bp1-barometric-pressure/). The HMP 155 humidity and temperature probe is placed inside a standard radiation shield and at 3.2 m from the ground in order to avoid that the snowpack eventually covers the sensors.

Along the five snow seasons analyzed, the mean annual air temperature ranged between 5.26ºC (2014/15 snow season) and 3.51 ºC (2012/13 snow season), with an average





value of 4.63 ℃. The accumulation period has shown a mean air temperature that ranged from -2.78℃ (2012/13) to -1.15 ℃ (2011/12 snow season) being -1.79 ℃ the average value for the whole study period. Finally, the melting period showed a mean value of 5.16 ℃ ranging from 2.79 ℃ (2012/13 snow season) to 7.30 ℃ (2014/15 snow season). Table 1 shows the 2012/13 snow season was the coldest one of the study period. Figure 4 depicts the temporal evolution of air temperature and other variables observed in the AWS. Thus this figure shows the control of air temperature on ground and surface temperature.

The relative air humidity and the atmospheric air pressure are shown in Table 2 and 3, respectively. The mean annual value of the relative humidity for the five seasons is 65%, with a 71% during the accumulation period and 69% during the melting one. Similarly, atmospheric air pressure has a mean annual value of 791 mb, being for the accumulation period 789 mb and for the melting period 788 mb.

### 3.3 Ground temperature

On 22 November 2012 four Campbell Scientific "107 temperature probes" (https://s.campbellsci.com/documents/es/manuals/107.pdf ) were installed in the AWS to measure ground temperature at different depths. One sensor was located in the atmosphere-ground interface (slightly buried, 0 cm depth), while the other three were respectively placed at 5 cm, 10 cm and 20 cm depths. Table 4 and Figure 4 show the average values of ground temperatures and the temporal evolution of ground temperature. There exists a lack of data from august 2016 onwards because temperature probes were damaged by cows. The average values during the period with information for the 0 cm, 5 cm, 10 cm and 20 cm depths are respectively: 5.26±6.22 ℃, 4.97±5.52 ℃, 4.93±6.17 ℃, 4.89±4.56 ℃.

The temporal evolution of air and ground temperatures depicts the impact of snowpack presence on ground energy dynamics. The snowpack shelters ground from the high temporal variability of air temperature. Therefore, ground temperatures have a significant smooth in the daily variability. Additionally, it is observed how the different ground temperatures tend to reach 0℃ while snow covers the ground; i.e., the typical soil-snow interface temperature.

### 3.4. Surface temperature

At the same date of the installation of the ground temperature sensors, an IR100 infrared remote temperature sensor (Campbell Scientific, https://s.campbellsci.com/documents/eu/manuals/ir100_ir120.pdf ) was also set up to




measure surface temperature of near target ground or snow. On Table 5 is shown the average land surface temperatures. The mean annual surface temperature is 2.56ºC, with a mean value of -4.81ºC during the accumulation period and 3.27ºC during the melting period.

The infrared remote sensor shows the snow surface tendency to cooling faster than soil. During winter and spring, while snow is present on the ground, air temperature and surface temperature shows higher differences, being always observed lower surface temperatures (see the occurrence of snow below the AWS when lower surface temperatures are observed in Figure 4). This is plainly exemplifying the higher energy

irradiance of snow when compared to free snow soils.

**3.5 Global and reflected solar irradiance**

The AWS also obtains information on the global and reflected solar irradiance with a CMA 6 Kipp&Konen albedometer (http://www.kippzonen.com/Download/72/Manual-Pyranometers-CMP-series-English). Figure 4 shows the daily evolution of the values

recorded, and how these are interrelated increasing the reflected radiation when incident does. The average values of these variables are presented in Table 6. For the whole period, the average values of the incident radiation are 207.97 W/m$^2$day considering complete snow seasons, 161.15 W/m$^2$ day accounting accumulation periods and 276.93 W/m$^2$day considering all melting period. Similarly, the reflected radiation average

values are: 83.67 W/m$^2$day for entire snow seasons, 109.57 W/m$^2$day for the accumulation periods and 119.57 W/m$^2$day for melting periods.

Similarly to ground and surface temperatures, the radiation reflected is markedly influenced by snow presence. Periods in which snow is present over the ground the sensor show higher values of reflected radiation when compared to snow free periods

(Figure 4).

**3.6 Snow depth and precipitation**

The AWS is also equipped with a sonic ranging sensor (Campbell Scientific SR50A, https://s.campbellsci.com/documents/cr/manuals/sr50a.pdf ). For simplicity we will refer it as snow depth sensor since it is used for measuring how evolves the distance

between the surface and the sensor (the sensor is placed 2.64 m from the ground being obtained snow depth subtracting to his value the observed distance). This sensor has worked without any interruption during the study period and provides a good climatology of the snow depth evolution in the Izas Experimental Catchment. Therefore, the information of the snow depth can be used as a reference for other



observations of snowpack evolution. The average values for the whole study period are: 89.20 cm for the accumulation period and 53.32 cm for the melting period (Table 7 shows the seasonal values). The temporal evolution of the snow depth is shown in Figure 4.

5  Additionally Figure 4 shows the precipitation values for the period with consistent data in the precipitation gauge (since end July 2014). The sensor installed is a Geonor T-200B with wind shield (http://www.geonor.com/brochures/t-200b-series-all-weather.pdf), which continuously weights the accumulated precipitation (liquid and solid). The precipitation accumulated over a certain period is calculated subtracting final and initial weighted values. Table 7 includes the accumulated precipitation for the whole snow year and also during the accumulation and melting periods.

## 4. Information on snow distribution

### 4.1. TLS acquisitions of snow depth distribution

During the five snow seasons presented here, from three to six TLS surveys were accomplished each year in the Izas Experimental Catchment. TLS are devices that use LiDAR technologies, a remote sensing method that obtains the distance between a target area and the device. During a TLS data acquisition, the device measures the distance of some hundreds of thousands of points within the area defined by the

operator, creating a cloud of data points representing the topography of the target surface. The device used in this study is a long-range TLS (RIEGL LPM-321 (Fig.2), http://www.riegl.com/uploads/tx_pxprieg1downloads/10_DataSheet_LPM-321_18-03-2010.pdf ). The technical characteristics of this model are: (i) light pulses of 905 nm wavelength (near-infrared), appropriate for acquiring data from snow cover (Prokop,

2008); (ii) a minimum angular step width of 0.018°; (iii) a laser beam divergence of 0.046°; and (iv) a maximum working distance of 6000 m. In order to reduce topographic shadowing (note that terrain topography limits the line-of-sight of the TLS) two scanning positions (Scan station on Fig.1) were established within the study site (Figure 1). Additionally 12 reflected targets were fixed on the terrain (Fig. 2). The

location of these targets was acquired on each TLS acquisition date, since this information is used in the post-processing phase for comparing the point clouds acquired in the different dates. The protocol for obtaining the information in the field and the methodology for generating the snow depth distribution maps for the different



TLS survey dates is fully explained in Revuelto et al., 2014a. The method is mainly based on calculating the elevation difference between the point clouds obtained on different dates with and without snow presence over the study area. The final products are snow depth distribution maps with grid size of 1x1 m, with a mean absolute error of

0.07 m in the obtained snow depth values (Revuelto et al., 2014a).

Figure 5 shows the snow depth maps obtained for the 2012/13 snow season. It is presented the information of this snow season because six TLS surveys were achieved. Additionally the accumulated snow depths were quite important and thus reproduce an interesting example of snowpack evolution on time. These maps show the high spatial

variability of the snowpack within the study area, with marked changes in the snow depth distribution in short distances. Also it is observed how high accumulation areas have important accumulations during the whole snow season with a thick snowpack for dates in which the snow cover have completely melted over large areas of the catchment.

Table 8 presents the average snow depth and the maximum snow depth value observed for each TLS acquisition. It is also shown in this table the coefficient of variation on each snow distribution map and also the fraction of the snow covered area. The values obtained depict the important snow depth accumulation occurring in some areas of the catchment while the average snow depth is lower.

**4.2. Snow covered area from time-lapse photographs**

The Izas Experimental Catchment is also equipped with a Campbell CC640 digital camera (https://s.campbellsci.com/documents/sp/manuals/cc640.pdf ). This camera was mounted with a solid metal structure fixed in the terrain with concrete (Figure 2). Hereby it is ensured a constant position that gives consistency to the information

obtained. The digital camera has a resolution of 640x480 pixels with a focal length of 6-12 mm. The field of view of the photographs obtained with the camera mounted in the metal structure cover approximately 30 ha (Figure 1), what represents about a 52% of the total surface covered by the TLS. The camera obtains three pictures per day (time-lapse photography) at 10:00, 11:00 and 12:00 UTC, ensuring a good illumination of the

area. Figure 6 show four photographs obtained during the 2012/2013 snow season, in which can be observed how snow covered area evolve in time.

The pictures obtained can be projected into a Digital Elevation Model (DEM) of the study site. Projecting the pictures into the DEM along an entire snow season provides distributed information on the snow covered area evolution in the same reference



system of snow depth maps. The approach for projecting the pictures into the DEM is described by (Corripio, 2004) and the specific features of the methodology applied in the Izas Experimental Catchment are fully described in (Revuelto et al., 2016a). The routines applied does, in first term a viewing transformation considering the optics of

the camera and in second term a perspective projection, providing a virtual image of the DEM. Therefore, in the second step the correspondence of ground control points within the surveyed area with pixels of the photograph must be established. Since this stage is quite sensible, the coordinates of ground control points were acquired with a differential GPS. Finally, the daily series of the projected images can be definitely binarized to

create daily snow presence/absence maps. This information can also be used for other application as for example to observe the growth timing of plant species.

Since the binarized snow presence/absence maps have almost a daily frequency (note that about a 20% of all photographs from the camera had to be discarded because cloud or snow presence), many parameters can be derived from this information, including the

Snow Covered Area temporal evolution, the numbers of days with snow presence or the melt out date (MOD) on each pixel. Figure 7 shows an example of the number of days with snow presence for the 2011/12 and 2012/13 snow season.

### 5. Data availability

The database presented and described in this article is available for download at Zenodo (https://doi.org/10.5281/zenodo.579979). Meteorological data of the AWS are ready in .csv format. The meteorological dataset includes observations in 10 min interval. For an easier transferability and also to allow a wider post-processing the TLS survey point clouds containing the observed snow depth distribution are available on-line (one file

for each TLS acquisition). These point clouds are in UTM 30T North coordinate system. Cloud free days photographs of the time-lapse camera are available in the online repository with the correspondence of pixel-ground control points GPS coordinates. Also it is provided information on the optics and chip size of the camera.

### 6. Summary

The Izas Experimental Catchment is a well-established study area in the south face of the Pyrenees in which different meteorological and snow variables are automatically acquired. Additionally, an important effort on field data acquisition with TLS has been conducted during the last five snow seasons and is still maintained. The dataset



described here is novel in the Pyrenees because it represents for the first time high spatial resolution information on the snowpack distribution and its evolution on time being also available continuous information on meteorological variables. The high quality of the information obtained has been already exploited for different studies on

the understanding of snowpack dynamics and on the improvement of simulation approaches of snowpack evolution in mountain areas (López-Moreno et al., 2012, 2014, Revuelto et al., 2014b, 2016a, 2016b). However, there exist many scientific questions still unanswered as the long term influence of topography on snow dynamics, the spatial distribution of snow during precipitation and wind blowing events. Also, the high inter-

annual variability of snow accumulation in the Pyrenees has important consequences for water management, especially in the Mediterranean area (García-Ruiz et al., 2011). Thus, it is quite important to continue obtaining information on snowpack evolution and on the meteorological variables that control snow dynamics. This information will allow to the scientific community to better understand processes involved allowing a better

adaptation to climate change scenarios. Moreover, offering the possibility of exploiting the information to other colleges provides the opportunity of establishing new collaboration networks to push forward the science limits in mountain areas.

**7. Acknowledgments:**

This study was funded by the research projects CGL2014-52599-P "*Estudio del manto de nieve en la montaña española y su respuesta a la variabilidad y cambio climatico*" (Ministry of Economy and Development, MINECO) and CLIMPY: "*Characterization of the evolution of climate and provision of information for adaptation in the Pyrenees*" (FEDER-POCTEFA). The authors thanks the unique opportunity of sharing information

thought the International Network for Alpine Research Cathcment Hydrology (INARCH) .J. Revuelto is supported by a post-doctoral Fellowship of the AXA research found 2016 call (*le Post-Doctorant Jesús Revuelto est bénéficiaire d'une bourse postdoctorale du Fonds AXA pour la Recherche*). C. Azorin-Molina is supported by the Marie Skłodowska-Curie Individual Fellowship (STILLING project – 703733) funded

by the European Commission



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





## Figures

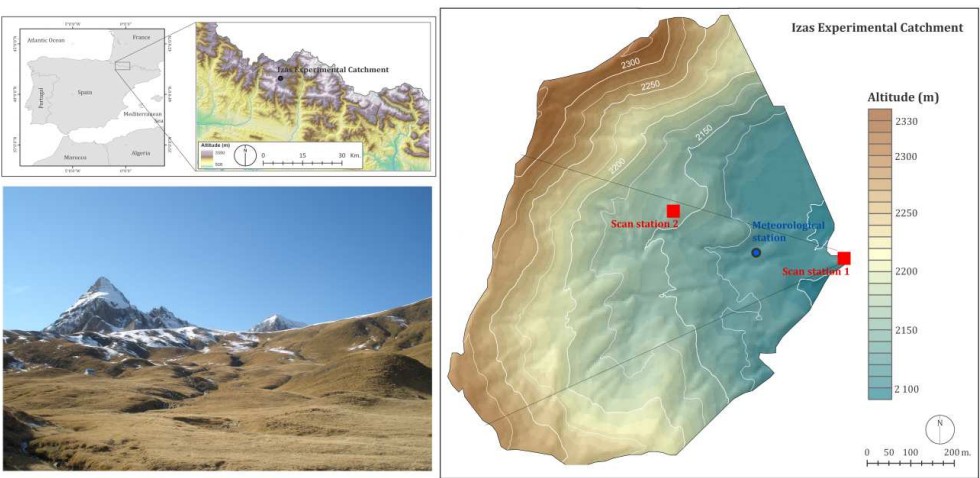

**Figure 1:** The Izas Experimental Catchment study site. Upper left figure shows the location of the study site. In the lower left panel, it is shown an overview picture of the
5   catchment with marginal snow presence. The right map shows the topographic characteristics of the catchment and the location of the TLS scanning positions (Scan stations), the meteorological station and the field of view of the time-lapse camera (continuous lines from Scan station 1).

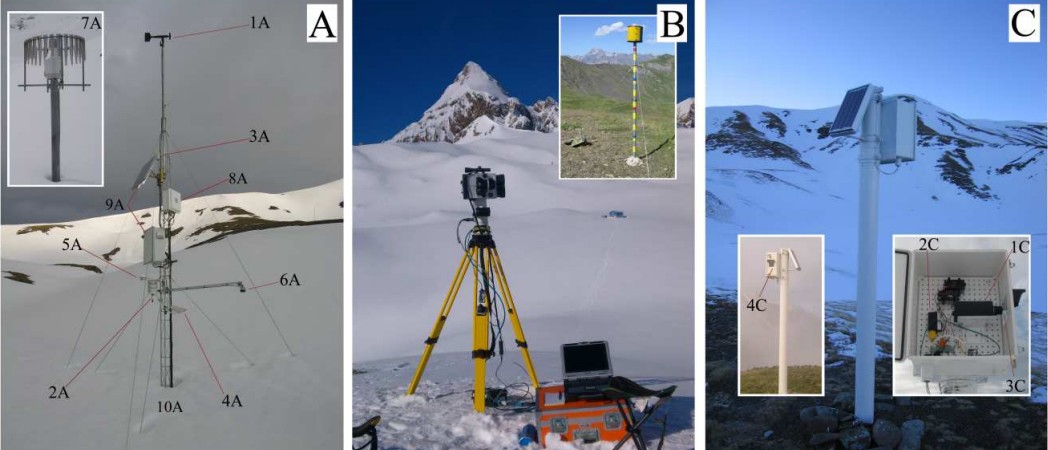

**Figure 2:** Pictures of the experimental site equipment. (A): AWS sensors. 1A: Young Wind Sensor, 2A: Radiation shield with HMP 155 humidity and temperature probe, 3A BP1 air pressure recorder, 4A: IR100 infrared remote temperature sensor, 5A: CMA6 Kipp & Konen albedometer, 6A: SR50A range sensor, 7A: Geonor T-200B with wind shield, 8A: CR3000 datalogger and modem, 9A: Solar panel and battery, 10A: Campbell Scientific 107 ground temperature probes. (B) RIEGL LPM-321 TLS mounted in the tripod during an acquisition campaign. In the upper-right part it is shown one of the 12 fixed reflective targets fixed on the terrain. (C) Campbell CC640 camera mounted in the metal structure. 1C: digital camera inside the enclosure house, 2C: modem, 3C: protection glass of the digital camera, 4C: frontal view of the camera and its structure.





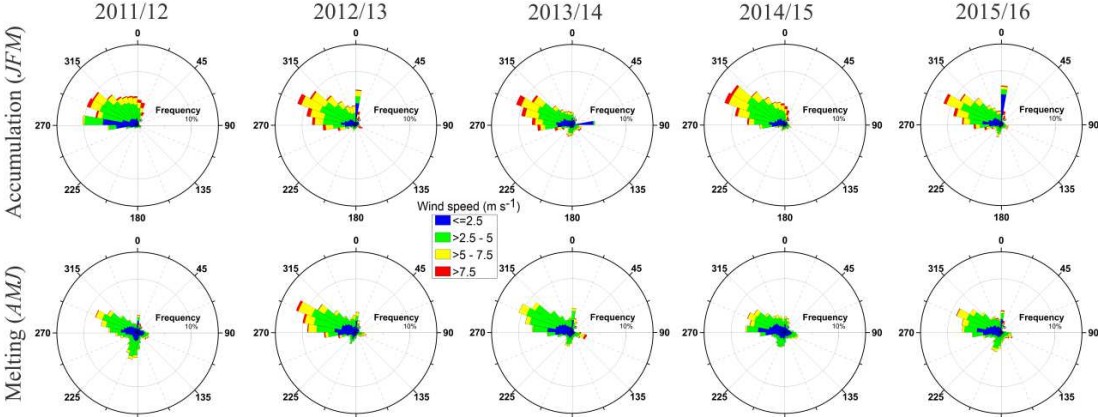

**Figure 3**: Wind roses showing the frequency (in %) of wind speed and direction observed in the AWS for accumulation (upper wind roses) and melting (lower wind roses) snow seasons.





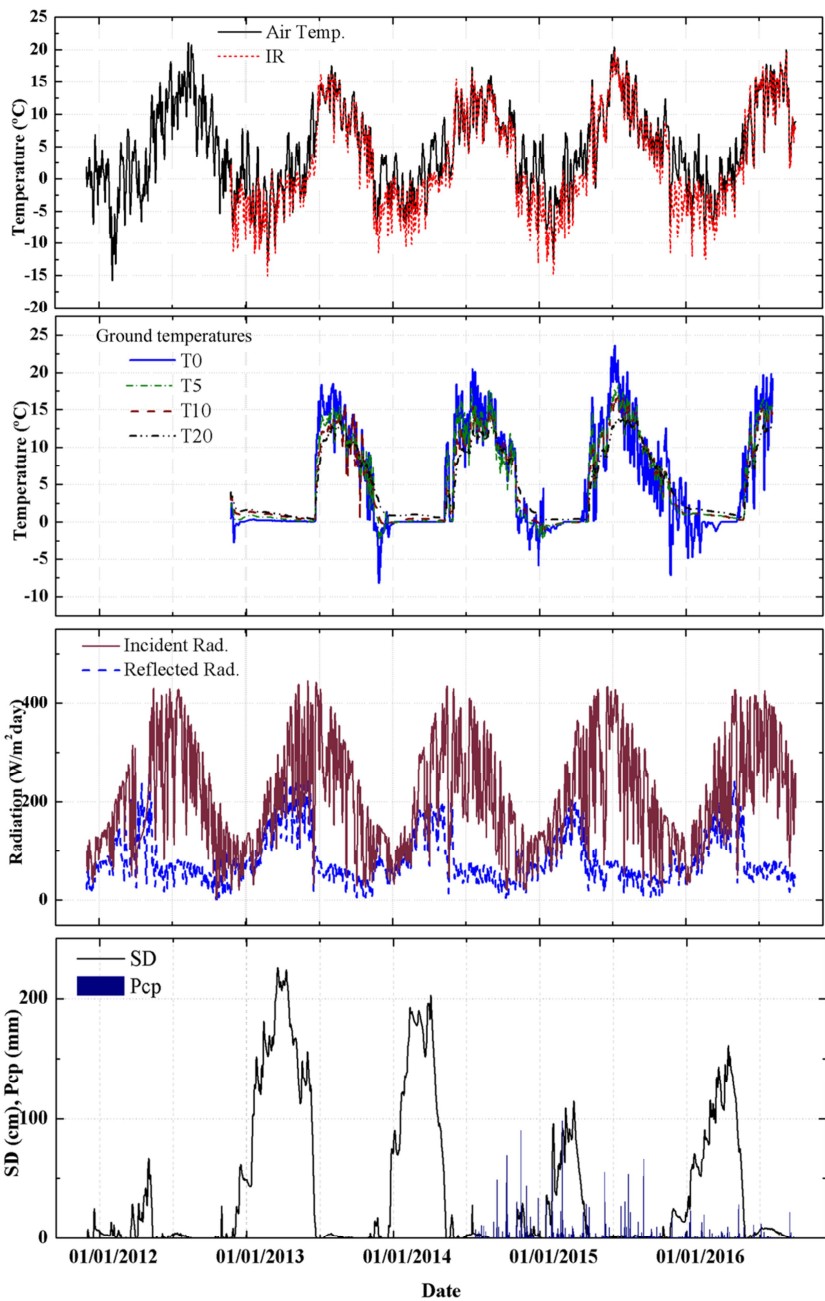

**Figure 4**: Temporal evolution of meteorological variables during the study period. From up panel to the low panel is shown air temperature and surface temperature (from the IR sensor); ground temperature for the four depths; global (Incident) and reflected solar irradiance; and punctual snow depth (SD) and daily accumulated precipitation (Pcp) (sum of solid and liquid).



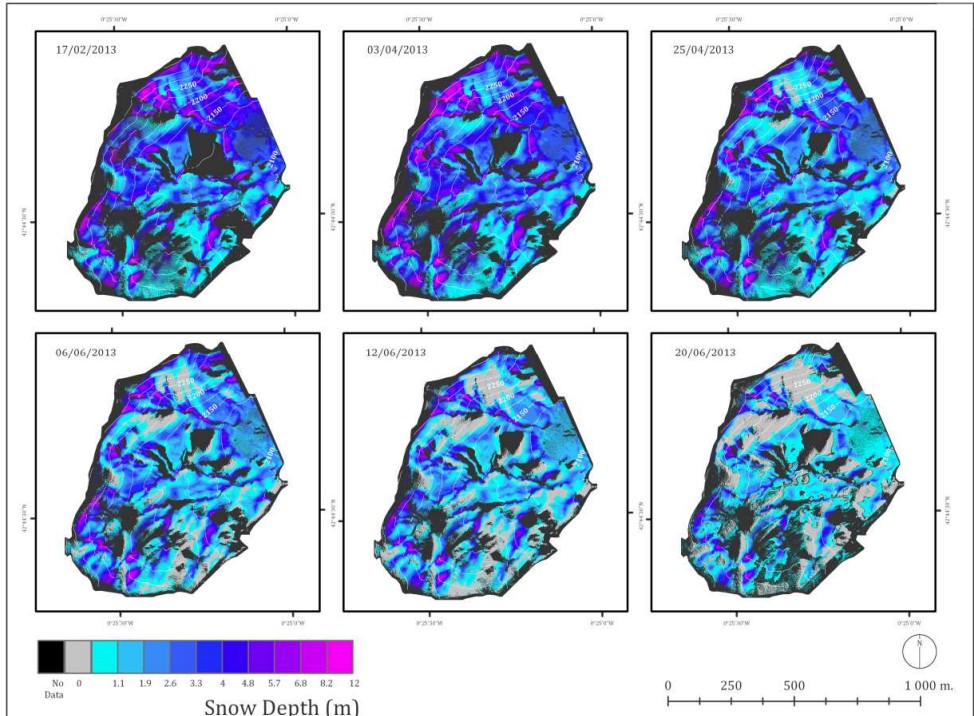

**Figure 5** Snow depth distribution maps obtained for the six TLS acquisitions dates of 2012/2013 snow season.



.

**Figure 6** Example of a sequence of four photographs for the 2012/13 snow season,
showing the snow covered area evolution.



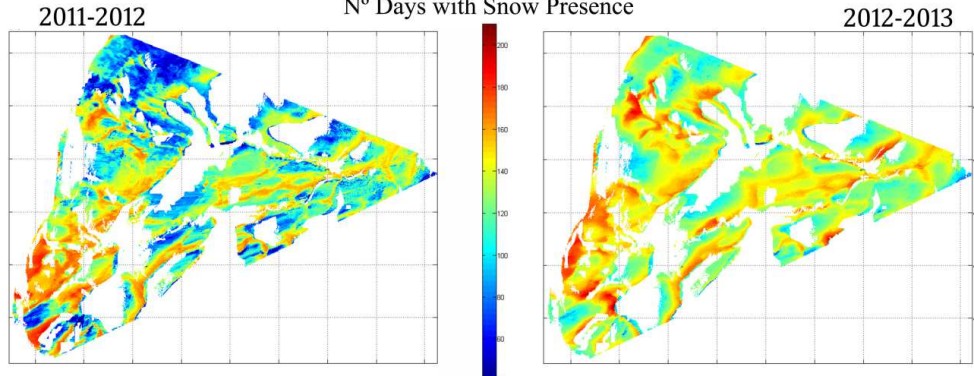

**Figure 7** Number of days with snow presence for each pixel for 2011/12 and 2012/13 snow seasons.





**Tables:**

| | | Air temperature (ºC) | | | | |
|---|---|---|---|---|---|---|
| | | 2011/12 | 2012/13 | 2013/14 | 2014/15 | 2015/16 |
| **Mean** | **Annual** | 5.13±7.73 | 3.50±6.88 | 4.17±6.11 | 5.26±7.02 | 5.08±6.69 |
| | **Accumulation** | -1.15±5.69 | -2.78±4.57 | -1.71±3.44 | -1.65±4.87 | -1.66±3.69 |
| | **Melting** | 5.80±6.60 | 2.79±4.79 | 5.51±4.07 | 7.23±4.86 | 4.45±5.12 |
| **Max** | **Annual** | 25.87 | 20.85 | 21.42 | 24.07 | 24.23 |
| | **Accumulation** | 7.89 | 10.69 | 10.20 | 10.98 | 11.62 |
| | **Melting** | 18.29 | 17.13 | 18.32 | 23.07 | 19.26 |
| **Min** | **Annual** | -18.51 | -15.26 | -11.35 | -15.24 | -11.78 |
| | **Accumulation** | -18.51 | -15.26 | -11.35 | -15.24 | -11.78 |
| | **Melting** | -9.33 | -9.04 | -3.71 | -4.76 | -8.20 |

**Table 1:** Mean and standard deviation of air temperature for the five snow seasons for the annual, accumulation and melting periods. Also are shown maximum and minimum air temperatures for the each period of the snow seasons.

| | Relative Air Humidity (%) | | | | |
|---|---|---|---|---|---|
| | 2011/12 | 2012/13 | 2013/14 | 2014/15 | 2015/16 |
| **Annual** | 59.9±18.9 | 70.1±17.1 | 68.8±17.3 | 64.8±19.2 | 65.9±18.5 |
| **Accumulation** | 67.1±18.1 | 70.5±19.3 | 72.7±15.8 | 62.8±22.2 | 71.3±18.3 |
| **Melting** | 57.1±15.2 | 74.4±14.5 | 68.7±15.9 | 63.9±15.8 | 69.9±14.1 |

**Table 2:** Mean and standard deviation of relative humidity for the five snow seasons for the annual, accumulation and melting periods.

| | Atmospheric air pressure (mbar) | | | | |
|---|---|---|---|---|---|
| | 2011/12 | 2012/13 | 2013/14 | 2014/15 | 2015/16 |
| **Annual** | 794.5±5.9 | 790.7±7.7 | 791.3±6.5 | 792.4±6.9 | 791.8±7.1 |
| **Accumulation** | 790.9±7.2 | 784.7±8.3 | 786.4±6.9 | 789.7±9.3 | 786.8±7.9 |
| **Melting** | 797.1±3.6 | 790.9±6.6 | 791.8±4.6 | 794.2±4.4 | 788.9±5.4 |

**Table 3:** Mean and standard deviation of atmospheric air pressure for the five snow seasons for the annual, accumulation and melting periods.



|  | Depth (cm) | Ground Temperatures (ºC) | | | | |
|---|---|---|---|---|---|---|
|  |  | 2011/12 | 2012/13 | 2013/14 | 2014/15 | 2015/16 |
| **Annual** | **0** | Nan* | 4.60±6.71 | 5.13±6.45 | 5.98±7.02 | 4.24±6.02 |
|  | **5** | Nan | 4.35±5.67 | 5.61±6.52 | 6.06±5.52 | 4.66±5.12 |
|  | **10** | Nan | 4.38±5.19 | 5.07±5.46 | 5.99±6.09 | 4.55±4.87 |
|  | **20** | Nan | 4.26±4.66 | 5.01±4.62 | 5.08±3.26 | 4.51±3.88 |
| **Acc.** | **0** | Nan | 0,22±0,05 | 0.03±0.04 | -0.26±0.87 | -0.66±1.13 |
|  | **5** | Nan | 0,69±0.12 | 0.11±0.08 | -0.39±0.54 | 0.99±0.10 |
|  | **10** | Nan | 1.10±0.16 | 0.31±0.18 | -0.27±0.23 | 0.98±0.11 |
|  | **20** | Nan | 1.34±0.19 | 0.94±0.06 | 0.39±0.08 | 1.57±0.17 |
| **Melting** | **0** | Nan | 1.21±3.49 | 5.53±6.41 | 7.87±6.41 | 4.57±5.46 |
|  | **5** | Nan | 1.04±2.45 | 5.19 ±6.08 | 7.03±5.71 | 4.43±5.09 |
|  | **10** | Nan | 1.06±1.78 | 4.15±4.68 | 6.46±5.32 | 4.15±4.79 |
|  | **20** | Nan | 1.04±1.36 | 3.46±3.49 | 5.35±4.15 | 3.50±3.47 |

**Table 4:** Mean and standard deviation ground temperature for different depths for the five snow seasons for the annual, accumulation and melting periods (*Nan means no data observed during the period).

|  | Surface temperature (ºC) | | | | |
|---|---|---|---|---|---|
|  | 2011/12 | 2012/13 | 2013/14 | 2014/15 | 2015/16 |
| **Annual** | Nan | 1.29±7.83 | 2.44±7.06 | 3.26±8.14 | 3.26±7.71 |
| **Accumulation** | Nan | -5.38±3.58 | -4.18±2.65 | -5.36±3.61 | -4.32±2.99 |
| **Melting** | Nan | -0.09±3.44 | 3.75±5.16 | 5.95±6.02 | 3.47±5.96 |

**Table 5**: Mean surface temperature from the infrared sensor for the five snow seasons for the annual, accumulation and melting periods (*Nan means no data observed during the period).

|  |  | Radiation (W/m²day) | | | | |
|---|---|---|---|---|---|---|
|  |  | 2011/12 | 2012/13 | 2013/14 | 2014/15 | 2015/16 |
| **Annual** | **Global** | 219.48±110.60 | 205.36±114.50 | 196.64±110.49 | 207.63±116.50 | 211.03±113.95 |
|  | **Reflected** | 82.87±49.60 | 96.20±64.92 | 79.35±52.78 | 76.34±64.90 | 83.61±53.76 |
| **Acc.** | **Global** | 181.09±68.18 | 154.83±67.30 | 150.04±84.02 | 166.97±65.80 | 152.83±83.18 |
|  | **Reflected** | 99.14±40.34 | 117.04±44.35 | 108.50±47.43 | 114.24±44.35 | 108.94±48.59 |
| **Melting** | **Global** | 245.37±120.56 | 289.59±114.10 | 283.33±102.80 | 287.65±117.15 | 278.71±114.37 |
|  | **Reflected** | 103.11±67.15 | 169.56±60.28 | 114.83±61.10 | 90.51±60.28 | 120.06±67.30 |

**Table 6**: Mean global and reflected radiation for the five snow seasons for the annual, accumulation and melting periods.



|        |             | 2011/12      | 2012/13       | 2013/14       | 2014/15       | 2015/16       |
|--------|-------------|--------------|---------------|---------------|---------------|---------------|
| **Ann** | **Pcp (mm)** | Nan          | Nan           | Nan           | 1572          | 411           |
| **Acc.** | **SD (cm)**  | 14.74±14.60  | 145.61±52.3   | 148.54±41.60  | 55.90±36.50   | 81.42±31.67   |
|        | **Pcp (mm)** | Nan          | Nan           | Nan           | 454.35        | 147.22        |
| **Mlt.** | **SD (cm)**  | 1.60±1.57    | 131.42±64.64  | 51.57±64.95   | 12.70±22.24   | 64.30±59.85   |
|        | **Pcp (mm)** | Nan          | Nan           | Nan           | 249.61        | 121.05        |

**Table 7:** Accumulated precipitation (liquid and solid) for snow seasons with observations available. Average snow depth values for accumulation and melting periods for the five snow seasons (*Nan means no data observed during the period).



| Date | | Mean SD (m) | Max SD (m) | SCA (%) | CV |
|---|---|---|---|---|---|
| Snow season 2011/12 | 22-Feb | 0.46 | 5.53 | 67.2 | 1.35 |
| | 02-Apbr | 0.17 | 3.86 | 33.5 | 2.23 |
| | 17-Apr | 0.56 | 5.34 | 94.1 | 1.07 |
| | 02-May | 0.90 | 6.11 | 98.8 | 0.74 |
| | 14-May | 0.21 | 4.47 | 30.9 | 1.90 |
| | 24-May | 0.09 | 4.32 | 18.9 | 1.29 |
| Snow season 2012/13 | 17-Feb | 2.91 | 10.89 | 98.8 | 0.63 |
| | 03-Apr | 3.19 | 11.20 | 100 | 0.56 |
| | 25-Apr | 2.42 | 10.10 | 96.3 | 0.76 |
| | 06-Jun | 1.98 | 9.64 | 86.4 | 0.86 |
| | 12-Jun | 1.69 | 8.90 | 77.1 | 0.90 |
| | 20-Jun | 0.76 | 7.97 | 67.0 | 1.35 |
| Snow season 2013/14 | 03-Feb | 2.16 | 10.20 | 96.0 | 0.59 |
| | 22-Feb | 2.56 | 10.47 | 98.6 | 0.57 |
| | 09-Apr | 2.54 | 9.72 | 89.0 | 0.65 |
| | 05-May | 1.67 | 9.02 | 75.2 | 0.87 |
| Snow season 2014/15 | 06-Nov | 0.22 | 2.78 | 85.0 | 0.81 |
| | 26-Jan | 0.74 | 4.88 | 89.3 | 0.85 |
| | 06-Mar | 2.13 | 11.55 | 94.0 | 0.69 |
| | 12-May | 0.67 | 7.75 | 56.0 | 1.21 |
| Snow season 2015/16 | 04-Feb | 0.82 | 6.20 | 91.1 | 0.63 |
| | 25-Apr | 1.86 | 10.82 | 97.0 | 0.50 |
| | 26-May | 1.16 | 7.81 | 74.8 | 0.70 |

**Table 8:** Observed mean and maximum snow depth values, snow covered area (SCA, % of the total area covered by the TLS), and coefficient of variation for the observed snow distribution on the TLS survey dates.