# Peer review of "In situ observations of meteorological variables and snowpack"

_Earth System Science Data, 2017_

## Referee Comment (RC1) · Anonymous Referee #1 · 1 Jul 2017

General Comments

The manuscript by Revuelto et al presents a meteorological and remotely sensed snow cover dataset for the Izas Experimental Catchment, an alpine basin located in the Spanish Pyrenees, for the period 2011 to 2016. Snow cover measurements include spatially distributed snow depth obtained by terrestrial laser scanning and snow covered area over time by oblique terrestrial photography, which was projected orthogo-

nally to the terrain. Although the meteorological record length is relatively short and there are certain gaps and missing components (e.g. precipitation data are missing for most of the period, which is unfortunate), this represents an important contribution and the snow cover data are of high potential value for model development and testing work, and comparisons of snow accumulation and distribution patterns among different alpine environments. For this reason the manuscript should be published. However, there are several areas where the manuscript could be improved and further details would benefit the reader and users of the published data. Not least is the fact that the English is of rather poor quality and requires editing to meet the standards of this journal. There are number of major grammatical errors and sentences and phrases that are either confusing or incorrect (too many in fact for me to point out every instance), and thus I would recommend that the authors seek to have the manuscript edited by a language professional.

Specific Comments

Page 1. In the title and abstract, sub-alpine environments are referred to as "ambients" and the area of the site is referred to as its "extension". These are not the appropriate or best choice of words.

Page 1, line 21. There is no description of long-wave radiation measurements in the manuscript and no such data provided, other than the IR100 infrared surface temperature measurements.

Page 1, line 32-33. The phrase "till the date" is an example of a grammatical mistake that requires correction by a language professional.

Page 2, line6. I am confused by what is meant by "controlled by the timing of snow distribution". Perhaps this is something that could be clarified.

Page 2, line 28. The work "spam" is a typographical error.

Page 5, section 3. Perhaps the authors could include some more detail on the specific

conditions in the immediate vicinity of the meteorological station. For example, is the vegetation sparse and the site relatively open? Is the nearby terrain flat?

Page 5, line 28. Do the authors mean to say the gauge is located 15 m away from the AWS tower?

Page 6, line 1. I am confused by the expression "average the evolution". Does this mean that the conditions are representative? The temperature measurements are not the spatial average. This is likely a phrase that could be clarified with proper language editing.

Page 7, line 28. "significant smooth" is a grammatical error.

Page 8, section 3.5. What is the sensor height of the radiometer?

Page 8, line 30-31. The phrase "from the ground being obtained snow depth subtracting to his value the observed distance" does not make sense as written. The word "his" is a typo.

Page 9, line 5-10. What is the orifice height of the Geonor gauge?

Page 9-10, section 4.1. Is there any information on error assessment of the snow depths obtained by the terrestrial laser scanner for any (but ideally multiple, or all) of the acquisitions? This is quite important to be able have confidence across the range of measured depths, and at different distances from the scanner position. As it stands, this is an omission and there should be some description of how well this approach performed in the manuscript.

Page 10-11, section 4.2. Similar as for the TLS snow depth, it is important to include some type of error assessment for the positional accuracy of the re-projected photos for snow covered area measurement. Were there any independent ground control points (i.e. that were not used in the correction procedure of Corripio) that could be used to verify how well the imagery fit to the true locations over the landscape? This would be useful toward placing some error bounds around the snow covered area derived from

this imagery.

Page 11, line 14. Discarding due to "snow presence" is due to snow obscuring the camera as I understand it. Is this correct?

Page 12, line 25. "thought" is a typo; this should be "through".

Page 12, line 26. There is a period in front of the J for the first authors name.

Page 12, line 27. I think that "found" is a typo. Should this be "fund"?

Page 17, figure 1. The font on the figure inset map is illegibly small. Could this be made clearer by using larger font?

---

## Referee Comment (RC2) · Anonymous Referee #2 · 6 Jul 2017

This paper presents a set of 23 terrestrial laser scans of snow distribution patterns in the Izas Experimental Catchment in the Spanish Pyrenees acquired over 5 consecutive years. This dataset is ideal to study snowcover dynamics at very high spatial resolution. Data from a well-equipped weather station further enable complementary snow model simulations.

This dataset may fall short of the requirements for the special issue of ESSD, which are

listed as "at least 5 years of continuous data with hourly sampling intervals for meteorological data, daily precipitation and streamflow". Precipitation data is only available for 2 years, and streamflow data is missing entirely. In particular the lack of precipitation data for 16 out of 23 TLS datasets is very unfortunate. Nevertheless I suggest to accept the data assembled as sufficient and relevant enough for inclusion in this special issue. However, I urge the authors to add data from WY 2017 (if measurements were continued) so that at least three seasons of complete data become available.

Anyway, the paper requires a thorough revision. First of all, I second the opinion of the first reviewer who stated that the English requires significant editing and recommended that the authors seek help from a language professional. Furthermore, we are provided with data that are mostly unprocessed. While I personally prefer to work with raw data straight from the data logger, others may find negative values for snow depth or shortwave radiation disturbing. Either way, a critical review of the data quality should be mandatory. As an example, the record from the SR50 obviously shows vegetation growth in the summer (Figure 4: SD + IR), which could be misinterpreted as snowfall. Such issues have to be addressed in the paper. Finally, the data files need to be reworked. Important information is missing, the file headers contain errors and feature acronyms that require guesswork. In particular:

A) The TLS data is organized in different formats. Some files feature 6 columns, others only 4 columns. The header is not explained in either type of file. Obviously "C2M_signed_distances" is the snow height. But does the Z coordinate refer to the bottom or to the top of the snowpack? And how to interpret intensity readings?

B) A separate DTM file is missing (even though it could be derived from the TLS data if we knew what "Z" was).

C) Is there no runoff data available?

D) The AWS dataset features a column with the header "ALBEDO_avg", however the respective data is obviously reflected shortwave radiation, not albedo. What sampling

scheme was used to determine wind gust speed, if this is what WS_Max is supposed to be? And where is the longwave radiation data (c.f. P1/L21)?

E) Section 4.1 suggests TLS data to come as a 1x1m grid ("The final products are snow depth distribution maps with grid size of 1x1 m"). But this clearly is not the case. The same applies to Section 4.2 suggesting SCA data to be available as ortho-rectified grids. Again this is not the case. Speaking of which, I would appreciate if these processed datasets could become available as well.

Further comments / suggestions (P = page / L = line)

1) Title: Remove the second part of the title. This paper is not on the "importance of high quality data in sub-alpine ambients". And what are sub-alpine ambients?

2) Abstract: P3/L4-8 presents a nice summary of the datasets. A similar description should be given in the abstract too, starting P1/L19.

3) P1/L30: I would, somewhere in the introduction, expect a short overview about those studies to demonstrate what these datasets have already been used for.

4) P2/L17: I don't think the term "boreal" applies here .

5) P3/32: What exactly reaches 40%, a single front, the fronts in autumn, or the highest monthly averages?

6) P6/L6: Since this is a data paper you should describe your "automatic quality-control checks for removing outliers". Why are SR50 readings limited to values >= 0, while this is not the case for shortwave radiation data?

7) P9/L5 "consistent data" seems to imply that at least some data also exist for before July 2014. Please clarify.

8) P12/L17: You may want to mention INARCH here.

9) Links to webpages: Note that these links may become unavailable in a few years'

time. Consider a more permanent way to provide respective information.

---

## Author Comment (AC1) · 25 Aug 2017

Dear editor, dear reviewers,

We are pleased to submit a revised version of the manuscript. First of all we would like to thank you for your effort in improving the work. Your comments and recommendations have been very helpful and we think that they allowed us to come up with a revised manuscript. Below, we provide a point by point answer to all comments raised

[Figure]

in the reviews, and references to all changes that we have introduced in the revised manuscript. We provide the same document to both reviewers to facilitate their assessment of all our revisions.

Looking forward to your reply,

Jesús Revuelto and co-authors.

Author's comments:

First of all, and following reviewer's recommendation the manuscript has been edited by a language professional. The manuscript with all changes marked can be found in this discussion. Moreover major comments of reviewer 2 have been addressed as follows.

a) The database now includes data from WY2017 (until 31st July 2017) in which two more TLS acquisitions were performed and are also included in the data set. Thereby a new zenodo DOI is provided in the text for the download of the database (check supplementary material with the revised manuscript including this new reference).

b) Data quality has been reviewed. For instance the SR50 record now only presents snow depth values and vegetation growth has been removed.

c) Data files are now complete, with file headers without errors and the acronyms described in the metadata file included in the data set.

d) All TLS snow depth maps are now provided as ASCII files with a 1x1 m regular grid cell size. Also a snow free DEM of the study area with same grid cell size is available in the data set.

e) Unfortunately runoff data is not available in Izas experimental catchment.

f) The reviewer is right and we have changed "ALBEDO_avg" by "Reflected Solar radiation". Similarly, former "PYRA_Avg" column has been renamed as "Incoming Global solar radtion". The name of these two variables is now the same provided by the CMA

6 Kipp&Konen albedometer user manual. Regarding WS_Max (gust wind speed), our wind processing and recording systems comply with WMO requirements, establishing the gust duration in 3 seconds for each 10-minute time interval. There was no observation of long wave solar radiation so we have removed any reference to it in the text.

g) The database now includes all available 1x1 m grid snow depth maps from the TLS and melt out date distribution maps generated from time-lapse photography. Besides all good quality time-lapse images with snow presence within the study area are included in the dataset.

Specific comments/ suggestions:

Reviewer 1 (R1): In the title and abstract, sub-alpine environments are referred to as "ambients" and the area of the site is referred to as its "extension". These are not the appropriate or best choice of words.

Author's answer (A): "Ambient" has been changed by "environment" and "extension" by "area" in the manuscript.

R1: Page 1, line 21. There is no description of long-wave radiation measurements in the manuscript and no such data provided, other than the IR100 infrared surface temperature measurements.

A: References to long wave radiation have been removed from the text since these have not been obtained in the study area.

R1: Page 1, line 32-33. The phrase "till the date" is an example of a grammatical mistake that requires correction by a language professional.

A: This sentence has been removed.

R1: Page 2, line6. I am confused by what is meant by "controlled by the timing of snow distribution". Perhaps this is something that could be clarified.

A: This sentence now states: "directly controlled by the evolution of snow cover over time"

R1: Page 2, line 28. The work "spam" is a typographical error.

A: This typographical error has been corrected.

R1: Page 5, section 3. Perhaps the authors could include some more detail on the specific conditions in the immediate vicinity of the meteorological station. For example, is the vegetation sparse and the site relatively open? Is the nearby terrain flat?

A: More information is now included in the text: "located in a flat open area with sparse vegetation (mountain pastures)".

R1: Page 5, line 28. Do the authors mean to say the gauge is located 15 m away from the AWS tower?

A: Yes. The gauge is located 15 m away from the AWS.

R1: Page 6, line 1. I am confused by the expression "average the evolution". Does this mean that the conditions are representative? The temperature measurements are not the spatial average. This is likely a phrase that could be clarified with proper language editing.

A: This sentence has been corrected and now states: "the AWS records serve to describe the evolution of atmospheric variables occurring at the Izas Experimental Catchment."

R1: Page 7, line 28. "significant smooth" is a grammatical error.

A: This error has been corrected. Now this sentence states: "Therefore, the daily variability in ground temperatures is significantly lower"

R1 Page 8, section 3.5. What is the sensor height of the radiometer?

A: Sensor height is 3.4 m and now it is stated in the text.

R1: Page 8, line 30-31. The phrase "from the ground being obtained snow depth subtracting to his value the observed distance" does not make sense as written. The word "his" is a typo.

A: This error has been corrected.

R1: Page 9, line 5-10. What is the orifice height of the Geonor gauge?

A: The orifice height is 3.25 m. This information is now included in the text.

R1: Page 9-10, section 4.1. Is there any information on error assessment of the snow depths obtained by the terrestrial laser scanner for any (but ideally multiple, or all) of the acquisitions? This is quite important to be able have confidence across the range of measured depths, and at different distances from the scanner position. As it stands, this is an omission and there should be some description of how well this approach performed in the manuscript.

A1: This information was already included in the text as follows: "The final products are snow depth distribution maps with grid size of 1x1 m, with a mean absolute error of 0.07 m in the obtained snow depth values (Revuelto et al., 2014a)."

R1: Page 10-11, section 4.2. Similar as for the TLS snow depth, it is important to include some type of error assessment for the positional accuracy of the re-projected photos for snow covered area measurement. Were there any independent ground control points (i.e. that were not used in the correction procedure of Corripio) that could be used to verify how well the imagery fit to the true locations over the landscape? This would be useful toward placing some error bounds around the snow covered area derived from this imagery

A: Some information concerning this issue has been included in the text. However, since all ground control points have been used for projecting the pictures in the DEM, could not be used for assessing accuracy. The software provides the calibration performance of the transformation, which was of 3.3 pixels of the webcam images.

R1: Page 11, line 14. Discarding due to "snow presence" is due to snow obscuring the camera as I understand it. Is this correct?

A: Yes, it is correct and this information and it has been included in the text.

R1: Page 12, line 25. "thought" is a typo; this should be "through".

A: This typo error has been corrected.

R1: Page 12, line 26. There is a period in front of the J for the first authors name.

A: The period has been removed.

R1: Page 12, line 27. I think that "found" is a typo. Should this be "fund"?

A: Yes, it is a typo and it has been corrected.

R1: Page 17, figure 1. The font on the figure inset map is illegibly small. Could this be made clearer by using larger font?

A: Following reviewer's recommendation, the font of this figure has been increased.

Reviewer 2 (R2): Title: Remove the second part of the title. This paper is not on the "importance of high quality data in sub-alpine ambients". And what are sub-alpine ambients?

A: Manuscript tittle has been changed to: "Meteorological and snow distribution data in the Izas Experimental Catchment (Spanish Pyrenees) from 2011 to 2017.

R2: Abstract: P3/L4-8 presents a nice summary of the datasets. A similar description should be given in the abstract too, starting P1/L19.

A: The abstract now includes this description: The climatic dataset consists of (i) continuous meteorological variables acquired from an Automatic Weather Station (AWS), (ii) detailed information on snow depth distribution collected with a Terrestrial Laser Scanner (TLS, LiDAR technology) for certain dates across the snow season (between 3 and 6 TLS surveys per snow season) and (iii) time-lapse images showing the evolution of the Snow Covered Area (SCA).

R2: P1/L30: I would, somewhere in the introduction, expect a short overview about those studies to demonstrate what these datasets have already been used for.)

A: A short overview of this works is now included in the introduction.

R2: P2/L17: I don't think the term "boreal" applies here.

A: Boreal has been removed from the text.

R2: P3/32: What exactly reaches 40%, a single front, the fronts in autumn, or the highest monthly averages?

A: All fronts in autumn together reach a 40% of total annual precipitation. This has been specified in the text.

R2: P6/L6: Since this is a data paper you should describe your "automatic quality-control checks for removing outliers". Why are SR50 readings limited to values >= 0, while this is not the case for shortwave radiation data?

A: This quality-check has now also been applied for shortwave radiation data.

R2: P9/L5 "consistent data" seems to imply that at least some data also exist for before July 2014. Please clarify.

A: "Consistent" has been removed from the text to avoid misunderstandings.

R2: P12/L17: You may want to mention INARCH here.

A: INARCH is now mentioned in this section.

R2: Links to webpages: Note that these links may become unavailable in a few years' time. Consider a more permanent way to provide respective information.

A: References to user's manuals have been changed and now are included in the reference section.

Please also note the supplement to this comment:
https://www.earth-syst-sci-data-discuss.net/essd-2017-43/essd-2017-43-AC1-supplement.pdf

―――――――――――――――――

**Supplement:**

|    | Meteorological and snow distribution data in the Izas Experimental                                                                                                                                                                                                                                                                                                                                                                       |
|----|------------------------------------------------------------------------------------------------------------------------------------------------------------------------------------------------------------------------------------------------------------------------------------------------------------------------------------------------------------------------------------------------------------------------------------------|
|    | Catchment (Spanish Pyrenees) from 2011 to 2017                                                                                                                                                                                                                                                                                                                                                                                    |
|    | In situ observations of meteorological variables and snowpack                                                                                                                                                                                                                                                                                                                                                                            |
|    | distribution at the Izas Experimental Catchment (Spanish                                                                                                                                                                                                                                                                                                                                                                                 |
| 5  | Pyrences): The importance of high quality data in sub-alpine                                                                                                                                                                                                                                                                                                                                                                             |
|    | ambients.                                                                                                                                                                                                                                                                                                                                                                                                                                |
| ļ  | Jesús Revuelto 1,2 , Cesar Azorin-Molina 1,3 , Esteban Alonso-González 1 , Alba Sanmiguel-                                                                                                                                                                                                                                                                                                              |
|    | Vallelado 1 , Francisco Navarro-Serrano 1 , Ibai Rico 1,4 , Juan Ignacio López-Moreno 1                                                                                                                                                                                                                                                                                                      |
| 10 |  <li>1 Pyrenean Institute of Ecology, CSIC, Zaragoza, Spain</li> <li>2 Météo-France - CNRS, CNRM (UMR3589), Centre d'Etudes de la Neige, Grenoble, France</li> <li>3 Regional Climate Group, Department of Earth Sciences, University of Gothenburg,
Gothenburg, Sweden</li> <li>4 University of the Basque Country, Department of Geography, Prehistory and Archaeology,</li>  |
| 45 | Vitoria, Spain                                                                                                                                                                                                                                                                                                                                                                                                                           |
| 15 | Abstract: This work describes the snow and meteorological dataset available for the                                                                                                                                                                                                                                                                                                                                               |
| i  | Abstract: This work describes the show and incleorological dataset available for the
Izas Experimental Catchment in the Central Spanish Pyranees, from the 2011 to 20167                                                                                                                                                                                                                                                              |
|    | snow seasons. The experimental site is located onin the southern side of the Pyrenees                                                                                                                                                                                                                                                                                                                                                    |
|    | between 2000 and 2300 m above see level covering with an area extension of 55 ha                                                                                                                                                                                                                                                                                                                                                         |
| 20 | The site is a good example of a sub-alpine ambient environment in which the dynamics                                                                                                                                                                                                                                                                                                                                                     |
| 20 | of snow accumulation and melting dynamics have are of major importance in many                                                                                                                                                                                                                                                                                                                                                           |
|    | $\underline{or}$ show accumulation and meeting dynamics have $\underline{or}$ major importance in many mountain processes. The climatic dataset dataset consists of ( i ) continuous                                                                                                                                                                                                                                              |
|    | meteorological variables acquired from an Automatic Weather Station (AWS). (ii)                                                                                                                                                                                                                                                                                                                                                          |
|    | detailed information on snow depth distribution collected with a Terrestrial Laser                                                                                                                                                                                                                                                                                                                                                       |
| 25 | Scanner (TLS, LiDAR technology) for certain dates along across the snow season                                                                                                                                                                                                                                                                                                                                                           |
|    | (between 3 and 6 TLS surveys per snow season) and ( iii ) time-lapse images that                                                                                                                                                                                                                                                                                                                                                  |
|    | showing the evolution of the Snow Covered Area-evolution (SCA). The-includes                                                                                                                                                                                                                                                                                                                                                             |
|    | information on different meteorological variables acquired in the with an Automatic                                                                                                                                                                                                                                                                                                                                                      |
|    | Weather Station (AWS) such asare precipitation, air temperature, incoming and                                                                                                                                                                                                                                                                                                                                                            |
| 30 | reflected short solar radiation and long wave infrared surface temperature radiation,                                                                                                                                                                                                                                                                                                                                                    |
|    | relative humidity, wind speed and direction, atmospheric air pressure, surface                                                                                                                                                                                                                                                                                                                                                           |
|    | temperature (snow or soil surface) and soil temperature; all were taken of them at 10                                                                                                                                                                                                                                                                                                                                                    |
| ļ  | minute intervals. Snow depth distribution was measured during 23 field campaigns                                                                                                                                                                                                                                                                                                                                                         |
|    | using a Terrestrial Laser Scanner (TLS), and there is also available daily information                                                                                                                                                                                                                                                                                                                                                   |
| 35 | onof the Snow Covered Area (SCA) was also retrieved from time-lapse photography.                                                                                                                                                                                                                                                                                                                                                         |

**The**

data

(https://doi.org/10.5281/zenodo.848277https://doi.org/10.5281/zenodo.579979) is valuable since it provides high spatial resolution information on the snow depth and snow cover distribution, which is particularly useful when combinedin combination with meteorological variables to simulate the snow energy and mass balance. This information has already been analyzed in variousdifferent scientific studies onworks studying snow pack dynamics and its interaction with the local climatology or terrain topographical characteristics. However, the database generated till the date has great potential for understanding other environmental processes from a hydrometerological or

set

10 ecological perspective in which snow dynamics play a determinant role.

**1. Introduction**

The sSnowpack distribution and its temporal evolution have a marked influence onin many mountain processes. For instance, These include erosion rates and sediment transport (Colbeck et al., 1979; Lana-Renault et al., 2011), geomorphological and glaciological processes (López-Moreno et al., 2015; Serrano et al., 2001), and

- phenological cycles (Liston, 1999; Wipf et al., 2009) are directly controlled by the evolution on time timing of snow distribution cover over time. InOn the other hand, snow melting dynamics areis also of has major importance from a hydrological perspective since one-sixth of thetotal Earth's total population depends on the water storage in mountain rivers headwaters (Barnett et al., 2005). In downstream areas exposed to extreme climatic conditions, the snowmelt runoff from mountain areas becomes a key element -(Viviroli et al., 2007), especially in these-zones affected by water shortages.subjected to water scarcity. Such This is the case of semi-arid regions, likeas the Mediterranean area, which is characterized by an irregular climate with long
- drought periods (Vicente-Serrano, 2006), and therefore by its dependence it is highly dependent on water stored in mountain areas, such as the Pyrenees-is quite high (López-Moreno, 2005; López-Moreno et al., 2008).

The Pyrenees are a mid-latitude mountain range, with significant snowfalls in the-more elevated presence in high elevation areas throughoutalong the year. During the boreal

- 20 spring, Pyrenean river discharges depend on the snow-melt melting of snowtiming, directly accounting from snow aboutwith approximately 40% of spring runoff being directly attributable to snow (López-Moreno and García-Ruiz, 2004). Thus, snow accumulation has a heavylarge influence on Pyrenean headwaters. This dependence is mostly<del>rather</del> due to the generally continuous snow cover from November to April above
- 2000 m above sea level (a.s.l.) (Alvera and Garcia-Ruiz, 2000; García-Ruiz et al., 1986; 25 López-Moreno et al., 2001). This way and, therefore, the study of the snowpack atim high elevations-areas of in the Pyrenees is crucial for understanding and managing mountain river discharges (López-Moreno, 2005), especially in the scenario offrame of a global climate change-scenario (García-Ruiz et al., 2011). However, the existence of
- 30

continuous snow observations above 2000 m a.s.l. is scarce in this mountain range, sincebeing most of them only have informationavailable from 1600 to 2000 m a.s.l. and when available these observations spam span those that are available only cover short time spans.<del>periods.</del> Therefore, by well-established study areas atim high elevations with,

10

5

having continuous measurements of meteorological variables and snowpack distribution are required in the Pyrenees.

In tThis paper, it is presentsed the recently acquired dataset of meteorological and snowpack variables obtained from a small size experimental catchment onof the

- 5 southern faceside of the Pyrenees. Although meteorological and hydrological data are available fromsince previous years (some variables have beenwere measured since the late 1980s<del>80's</del> (Alvera and Garcia-Ruiz, 2000)), we presentoffer data from the 2011/12 to 20152016/176 snow seasons, as data series provide higher quality and continuity, and also they-match with-in situ observations of snow depth and snow cover.
- 10 The dataset consists of (i) continuous meteorological variables acquired from an Automatic Weather Station (AWS), (ii) detailed information on snow depth distribution collected with a Terrestrial Laser Scanner (TLS, LiDAR technology) for certain dates acrossalong the snow season (between 3-2 and 6 TLS surveys per snow season) and (iii) time-lapse images that showing the Snow Covered Area evolution (SCA). Some years
- 15 of this dataset haves already been used to study the topographic control on snow depth distribution (Revuelto et al., 2014b), the spatial variability of snow-pack at different distances (López.Moreno et al., 2012) or to investigateing how detailed snowpack simulation could be improved by including snow distribution information (Revuelto et al., 2016a,b).
- 20 The paper is structured as follows: Section 2 describes the study area characteristics; Section 3 presents meteorological data acquired from the AWS with a general description of the observed climatology; Section 4 describes the distributed measurements on snow depth distribution from the TLS and the SCA derived from time-lapse images; Section 5 concludes with information for downloading the database;
- and finally Section 6 summarizes all information available and the potential application of the database

**2. Study area characteristics and climatology**

**2.1. The Pyrenees**

30 The Pyrenees lies onin the northeastern borderlimit of the Iberian Peninsula (Figure 1) and form. It is an orographic barrier between the north and south faces. This way a Due to this, progressively higher aridity is found toward the southsouthward as a consequence of the mountain range blocksing humid air masses from the Atlantic

(López-Moreno and Vicente-Serrano, 2007; Vicente-Serrano, 2005). Thus, the natural barrier directly influences precipitation, leading toand as a consequence areas above 2000 m a.s.l. receivinge about 2000 mm/year, increasing to 2500 mm/year- in the highest divides of the mountain range and rapidly decreasinge to 600-800 mm/year in low elevation areas onf the southern side (García-Ruiz, et al., 2001).

5

Another distinct feature of the Pyrenees is their location between two water masses with contrastinged conditions; i.e., in the western side is the Atlantic Ocean is on the west side, while in the east side lays the Mediterranean Sea lies in the east. This positionsituation between both water masses causesoriginates a climatic transition from 10 Oceanic to Mediterranean conditions into the east. During autumn, fronts approaching from the Atlantic bring the highest monthly averages of precipitation in the western observatories, reaching the with their total contribution accounting forof all these fronts 40% of total annual precipitation in this area (Creus-Novau, 1983). <del>a</del> OppositelyConversely, spring and summer storms mostly affect the eastern areas of the Pyrenees, being favored promoted by the development of zones where sea breezes and 15 local winds convergence zones that to initiate deep moist convection along the eastern fringe of the Iberian Mediterranean area (Azorin-Molina et al., 2015). Therefore, by Pyrenean observatories in the east record a large numberhave a major contribution of

20

- convective events; e.g., reaching ai.e up to 32% of total annual precipitation in eastern valleys, but-and dropping below 16% of annual precipitation in western valleys (Cuadrat et al., 2007). In early winter, the arrival of fronts from the northwest and west are the most frequent, leading to the highest snow accumulation being found in the western Pyrenees (Navarro-Serrano and López-Moreno, 2017). The Azores high, which usually affects the Iberian Peninsula for at certain times in thesome winter periods,
- 25

originatesgives rise to relatively long periods with no snow accumulation in this season. Subsequently, in spring, snow accumulation isare associated with southwesterly advections, which lead to highheavy snow accumulations in the western Pyrenees (Revuelto et al., 2012). Snow remains for long periods above 1600 m a.s.l., between November and April (López-Moreno and Nogués-Bravo, 2006).

30 Similarly to precipitation, air temperature is influenced by the Atlantic-Mediterranean transitions, but elevation plays a major role inon its distribution. For instance, the lower annual thermal amplitude is-observed in the western Pyrenees is because of the proximity of the ocean-proximity (Cuadrat et al., 2007). As a general tendency in the

Central Pyrenees, the annual 0°C isotherms lielays between 2700 and 2900 m a.s.l. (del Barrio et al., 1990; Chueca, J., 1993).

Additionally the Pyrenees exhibit a high inter-annual variability in air temperature and precipitation, which makes involve great uncertainty in the annual snow accumulation

- very uncertain (López-Moreno, 2005). This variability is influenced by the inter-annual 5 variability of atmospheric circulation, being identified with -a decrease of snow accumulation weather types being identified under positive North Atlantic Oscillation (NAO) phases (López-Moreno and Vicente-Serrano, 2007). As observed with precipitation, snow accumulation correlates towith Atlantic-Mediterranean proximity and distance from to the main divide of the mountain range (Revuelto et al., 2012), and
- 10

is strongly dependent on the fluctuations of the 0°C isotherm during winter and spring. This high climatic variability is also the cause of originates a large inter-annual variability in total snow accumulation and on-its temporal distribution acrossalong the snow season (López-Moreno, 2005).

**2.2. The Izas Experimental Catchment 15**

The Izas Experimental Catchment (42°44' N, 0°25' W) has a surface area of 33 ha, but snow depth information covers a total of 55 ha, with elevations ranging between 2075 and 2325 m a.s.l. This area is close to the main divide of the Pyrenees in the headwaters of the Gállego River, near the Spain-France border (Figure 1). The Izas Experimental

20 Catchment exemplifies the general characteristics of sub-alpine areas of the Pyrenees. In this environment, snowpack dynamics are of have a major importance throughout along the year. Thus the atmosphere-snowpack interactions observed at this experimental site will enable ato better understanding of many processes inof sub-alpine areas.

The mean annual precipitation is 2000 mm, and snow accounts for approximately 50%

- of total precipitation (Anderton et al., 2004). For an average of 130 days each year the 25 mean daily air temperature is below 0 °C, with a mean annual air temperature of 3 °C, (del Barrio et al., 1997). Snow covers a high percentage of the catchment from November to the end of May (López-Moreno et al., 2010). Lithology shows limestones and sandstones of the Cretaceous period, and limestones of the Paleocene, much more
- resistant to erosion. The  $Z_z$  onal vegetation type corresponds to a high mountain steppe, 30 mainly covered by bunch grasses, namely Festuca eskia, Nardus stricta, Trifolium alpinum, Plantago alpine and Carex sempervirens. Rocky outcrops dominate in-the upper and steeper slopes (less than 15% of the study area). There are not trees present in the study area. The catchment is predominantly east-facing, with some areas also facing

north or south. The mean slope of the catchment is 16° (López-Moreno et al., 2012), with the topographic characteristics displayinghaving the typical high spatial heterogeneity on its topographic characteristics of sub-alpine areas, having with flat concave and convex areas.

5

10

**3. Meteorological data**

The study site is equipped with an AWS located in the lower elevation of the catchment (42° 44' 33.65''N, 0° 25' 8.83''W, 2113 m a.s.l., Figure 1), located in a flat open area with sparse vegetation (mountain pastures). The AWS measures wind speed and direction, atmospheric air temperature, relative humidity and air pressure, soil temperature for 0 cm, 5 cm, 10 cm and 20 cm, temperature of the surface close to the AWS (snow or soil, depending on whetherif snow is present or not), global and reflected solar irradiance, snow depth and precipitation (the precipitation gauge is

- located at 15 m of-from the AWS tower) (see Figure 2). Information on the main 15 atmospheric variables has been recorded since the end of 2011 (AWS installed on November 2011). Therefore, data availability covers five complete snow seasons. Since the station is located in the lower elevation of the catchment and despite air temperature lapse rate with elevation, the AWS records serve to describeaverage the evolution of atmospheric variables occurring at the Izas Experimental Catchment.
- 20 The data acquisition system consists of a Campbell Scientific CR3000 datalogger that samples each instrument and stores data at 10-minute time intervals. All data is transmitted via modem to the Pyrenean Institute of Ecology whereand once received we apply some automatic quality-control checks are applied to removefor removing outliers. Data gaps are rare for almost all variables and, therefore, instead of gap-filling
- with interpolation methods, only measured data are available. However, some variables 25 had long data gaps and certainthereby some periods have been discarded fromfor further analysis. This is the case of precipitation for the first three snow seasons, which were useless because of the length of data gaps.
- Since the main application of the data collected by the AWS is to assess the assessment of the snow cover evolution of snow cover in the study area, in the following 30 subsections we focus our analyses on the accumulation and melting periods: i.e., accumulation (January, February and March; JFM) and melting (April, May and June;

AMJ).-periods Annual values observed during a whole snow season are also presented for each sub-section.

**3.1. Wind speed and direction**

The AWS is equipped with a Young wind monitor - ALPINE MODEL Young

- Company, Model 05103-45-5, Wind Monitor -Alpine Model specifications © 5 2010)(Young Company, Model 05103-45-5; http://www.youngusa.com/Brochures/05103-45%20(0613).pdf), placed atin the highest point of the meteorological tower (8 m above the ground). The Pyrenees are commonly affected by strong westerly to northerly winds as shown in the wind roses displayed in Figure 3. With the exception of south winds 10 that mainly occurring during the melting period, westerly to northerly winds dominate.
  - Additionally, the the most frequently frequency of moderate to strong winds come from the north-west.. winds mainly occurs for northwesterly winds.

3.2. Air temperature, relative humidity and atmospheric air pressure

Air temperature and relative humidity wereare measured with the HMP155 Vaisala

- 15 sensor (Vaisala Company, HMP155 Humidity and Temperature Probe specifications © 2012http://www.vaisala.com/Vaisala%20Documents/Brochures%20and%20Datasheets/ HMP155-Datasheet-B210752EN-E-LoRes.pdf), and atmospheric air pressure\_was-is recorded with the BP1 sensor from Adcon telemetry (Adcon Telemetry Company, BP1 Pressure Sensor specifications, Barometric  $\bigcirc$ 2015
- 20 http://www.adcon.com/download/leaflet bp1 barometric pressure/). The HMP 155 humidity and temperature probe wasis placed inside a standard radiation shield and at 3.2 m from the ground in order to preventavoid that the snowpack from eventually coverings the sensors.
- AlongOver the five six snow seasons analyzed, the mean annual air temperature ranged between 5.26°C (2014/15-snow season) and 3.51-°C (2012/13-snow season), with an average value of 4.5963 °C. The accumulation period has shown a The mean air temperature in the accumulation period that ranged from -2.78°C (2012/13) to -0.56°C (2016/17)-1.15 °C (2011/12 snow season), being -1.79 °C the with an average value of -1.79°C for the whole study period. Finally, the melting period returned showed a mean 30 value of 5.516-°C ranging from 2.79-°C (2012/13-snow season) to 7.58°C (2016/17)to 7.30 °C (2014/15 snow season). Table 1 shows that the 2012/13 snow season was the coldest inone of the study period. Figure 4 depicts the temporal evolution of air

temperature and other variables observed in the AWS from 2011 to 2016. Thus, this

figure shows the control points forof air temperature on the ground and the surface temperature.

The relative air humidity and the atmospheric air pressure are shown in Tables 2 and 3, respectively. The mean annual value of the relative humidity for the five seasons is

65%, with a 7167% during the accumulation period, and 669% during the melt\_ing one. 5 Similarly, atmospheric air pressure has a mean annual value of 791 mb, with 7897 mbbeing for the accumulation period 789 mb and 79288 mb for the melt.for the melting period 788 mb.

**3.3 Ground temperature**

- 10 On 22 November 2012 four Campbell Scientific "107 temperature probes" (Campbell Ltd. 107 temperature Probe. Scientific C 2012https://s.campbellsci.com/documents/es/manuals/107.pdf ) were installed in the AWS to measure ground temperature at different depths. One sensor was located in the atmosphere-ground interface (slightly buried, 0 cm depth), while the other three were
- respectively placed at depths of 5 cm, 10 cm and 20 cm-depths. Table 4 and Figure 4 15 show the average values of ground temperatures and the temporal evolution of ground temperature. There exists a lack of dData is lacking from aAugust 2016 onwards because temperature probes were damaged by cows. The average values during the period with information for the 0 cm, 5 cm, 10 cm and 20 cm depths are respectively: 5.26±6.22 °C, 4.97±5.52 °C, 4.93±6.17 °C, 4.89±4.56 °C.
- 20

The temporal evolution of air and ground temperatures depicts the impact of the snowpack-presence on ground energy dynamics. The snowpack shelters ground from the high temporal variability of air temperature. Therefore, the daily variability in ground temperatures have a is significantly lowersmooth decrease in the daily variability. Additionally, it is observed how the Furthermore, the different ground temperatures tend to reach 0°C while snow covers the ground; i.e., the typical soil-snow

25

**30 3.4. Surface temperature**

interface temperature.

At the same date of Together with the installation of the ground temperature sensors, an IR100 infrared remote temperature sensor (Campbell Scientific Ltd, IR100/IR120 Infra-© 2015Campbell Scientific, sensor. remote temperature https://s.campbellsci.com/documents/eu/manuals/ir100\_ir120.pdf\_) was also set up to

measure surface temperature of near target ground or snow. On-Table 5 showsis shown the average land surface temperatures. The mean annual surface temperature is 2.56°C, with a mean value of  $-4.581^{\circ}$ C during the accumulation period and  $3.2794^{\circ}$ C during the melting period.

- The infrared remote sensor shows the tendency of the snow surface tendency to cooling 5 faster than soil. During winter and spring, while snow is present on the ground, the differences between air and surface temperature are more marked, with surface temperatures always observed to be lowerair temperature and surface temperature shows higher differences, being always observed lower surface temperatures (see the
- 10 occurrence of snow below the AWS when lower surface temperatures are observed in Figure 4). This is-plainly exemplifies ying the higher energy irradiance of snow when compared to free-snow-free soils.

**3.5 Global and reflected solar irradiance**

The AWS also obtains information on the global and reflected solar irradiance with a

- 15 CMA 6 Kipp&Konen albedometer Kipp & Zonen, CMP/CMA series pyranometer and albedometer ©. 2015) (http://www.kippzonen.com/Download/72/Manual-Pyranometers-CMP-series-English)placed at 3.4 m height. Figure 4 shows the daily evolution of the values recorded, and how these are interrelated, with increasing the reflected radiation increasing at the same time as the incident.when incident does. The
- average values of these variables are presented in Table 6. For the whole period, the 20 average values of the incident radiation are 207.97 W/m2day, taking complete snow seasons into account-considering complete snow seasons, 164.73161.15-W/m2 day afor<del>ccounting</del> accumulation-periods, and 280.95 276.93 W/m2day for<del>considering</del> all melting periods. Similarly, the reflected radiation average values are: 83.67 W/m2day for entire snow seasons, 109.69 109.57 W/m2day for the accumulation periods and

25

30

 $117.06 \frac{119.57}{100} W/m^2 day$  for melting periods.

Similarly to ground and surface temperatures, the radiation reflected is heavilymarkedly influenced by the presence of snow, presence. Periods in whichWhen snow coversis present over the ground, the sensor shows higher values of reflected radiation in comparison with when compared to snow--free periods (Figure 4).

**3.6 Snow depth and precipitation**

The AWS is also equipped with a Campbell SR50A sonic ranging sensor (Campbell Scientific SR50A, Campbell Scientific Ltd, SR50A Sonic Ranging Sensor, © 2011https://s.campbellsci.com/documents/cr/manuals/sr50a.pdf-). For the sake of Código de campo cambiado

simplicity we will refer to it as a snow depth sensor, since it is used for measuring how evolves the changing distance between the surface and the sensor (the sensor is placed 2.64 m from the ground, and the being obtained snow depth obtained by subtracting to his this value from the observed distance). This sensor has worked without any

- uninterruptedlyion during the study period and provideds a good elimatology-record of 5 the snow depth evolution in the Izas Experimental Catchment. Therefore, the information onf the snow depth can be used as a reference for other observations of snowpack evolution. The average values for the whole study period are: 89.2093.4 cm for the accumulation period and  $\frac{47.853.32}{2}$  cm for the melting period (Table 7 shows the 10 seasonal values). The temporal evolution of the snow depth is shown in Figure 4.
- In addition, Additionally Figure 4 shows the precipitation values for the period with consistent data in the precipitation gauge (since from the end of end July 2014). The sensor installed is a Geonor T-200B with wind shield (Geonor A/S, geonor T-200B series All-weather precipitation gauges, © 2010http://www.geonor.com/brochures/t-
- 200b-series-all-weather.pdf), which continuously weights the accumulated precipitation 15 (liquid and solid). The height of the gauge orifice is 3.25 m (2.5 m metal pedestrial plus the height of the T-200B inlet). The precipitation accumulated over a certain period wasis calculated by subtracting final and initial weighted values. Table 7 includes the accumulated precipitation for the whole snow year and also during the accumulation 20 and melting periods.

**4. Information on snow distribution**

**4.1. TLS acquisitions of snow depth distribution**

During the five snow seasons presented here, from three to six TLS surveys were 25 carried outaccomplished each year in the Izas Experimental Catchment. TLS are devices that useusing LiDAR technologyies, a remote sensing method that to obtain the distance between a target area and the device. During a TLS data acquisition, the device measures the distance of some hundreds of thousands of points within the area defined by the operator, creating a cloud of data points representing the topography of the target surface. The device used in this study is a long-range TLS (RIEGL LPM-321 (Fig.2), 30

RIEGL Laser Measurements, LPM-321 ©. 2010http://www.riegl.com/uploads/tx\_pxpriegldownloads/10\_DataSheet\_LPM\_321\_18\_ 03-2010.pdf). The technical characteristics of this model are: (i) light pulses of 905 nm

11

wavelength (near-infrared), appropriate for acquiring data from snow cover (Prokop, 2008); (ii) a minimum angular step width of  $0.018^{\circ}$ ; (iii) a laser beam divergence of  $0.046^{\circ}$ ; and (iv) a maximum working distance of 6000 m. In order to reduce topographic shadowing (note that terrain topography limits the line-of-sight of the TLS)

- two scanning positions (Scan station on Fig.1) were established within the study site
  (Figure 1). Additionally-12 reflected targets were also fixed onon the terrain (Fig. 2). The location of these targets was acquired on each TLS acquisition date, since this information is used in the post-processing phase for comparing the point clouds acquired onin the different dates. The protocol for obtaining the information in the field and the methodology for generating the snow depth distribution maps for the different TLS survey dates is fully explained in Revuelto et al., 2014a. The method is mainly based on calculating the elevation difference between the point clouds obtained on different dates with and without snow coverpresence acrossover the study area. The final products are snow depth distribution maps with a grid size of 1x1 m, with a mean absolute error of 0.07 m in the <del>obtained</del> snow depth values (Revuelto et al., 2014a).
  - Figure 5 shows the snow depth maps obtained for the 2012/13 snow season. It is presented t The information forof this snow season is presented because six TLS surveys were completed achieved. Furthermore, Additionally the accumulated snow depths were significant quite important and thus provide reproduce an interesting
- 20

25

example of snowpack evolution overon time. These maps show the high spatial variability of the snowpack within the study area, with marked changes in the snow depth distribution within short distances. Also it isIt was also observed how high accumulation areas hadwe largeimportant accumulations during the whole snow season, with a thick snowpack for dates onin which the snow cover had alreadywe completely melted over largewide areas of the catchment.

Table 8 presents the average snow depth and the maximum snow depth value observed for each TLS acquisition. It is also shown in tThis table also shows the coefficient of variation on each snow distribution map and also the fraction of the snow covered area. The values obtained depict the heavy accumulation of snow important snow depth

30 accumulation occurring in some areas of the catchment, while the average snow depth is lower.

**4.2. Snow covered area from time-lapse photographs**

The Izas Experimental Catchment is also equipped with a Campbell CC640 digital camera (Campbell Scientific Ltd, CC640 Digital Camera, ©

2010https://s.campbellsci.com/documents/sp/manuals/cc640.pdf). This camera was mounted onwith a solid metal structure set into the ground withfixed in the terrain with concrete (Figure 2)<del>.</del>, which Hereby it is ensured a constant position to obtain consistent information.that gives consistency to the information obtained. The digital camera has a

- 5 resolution of 640x480 pixels with a focal length of 6-12 mm. The field of view of the photographs obtained with the camera mounted onim the metal structure covers approximately 30 ha (Figure 1), whichat represents about a 52% of the total surface covered by the TLS. The camera obtaineds three pictures per day (time-lapse photography) at 10:00, 11:00 and 12:00 UTC, ensuring a-good illumination of the area.
- 10 Figure 6 contains four photographs fromobtained\_during the 2012/2013 snow season, in which can be observed howing how the snow covered area evolved in time. The pictures obtained can be projected into a Digital Elevation Model (DEM) of the study site. Projecting the pictures into the 1x1 m DEM foralong an entire snow season provides distributed information on the evolution of the snow covered area-evolution in
- 15 the same reference system asof snow depth maps. The approach for projecting the pictures into the DEM is described by (Corripio, 2004) and the specific features of the methodology applied in the Izas Experimental Catchment are fully described in (Revuelto et al., 2016a). The routines applied first makedoes, in first term a viewing transformation consideringallowing for the optics of the camera anda-in seconda-term a
- 20 perspective projection, providing a virtual image of the DEM. Therefore, in the second step1 the correspondence of ground control points within the surveyed area-with the pixels of the photograph must be established. Since this stage is quite sensitiveble, the coordinates of ground control points were acquired with a differential GPS. With this process, images projected-images into the DEM had a 3.3 pixel2s performance in the
- 25

calibration of the transformation. Finally, the daily series of the projected images can be definitely binarized to create daily snow presence/absence maps. This information can also be used for other applications, such as as for example to observe the growth timing of plant species.

30

Since the binarized snow presence/absence maps were recorded onhave almost a daily frequency (note that about a 20% of all photographs from the camera had to be discarded because cloud or snowpresence\_obscureding the camera lens), many parameters can be derived from this information, including the Snow Covered Area temporal evolution, the numbers of days with snow presence or the melt out date (MOD) on each pixel. Figure 7 shows an example of the number of days with snow presence for the 2011/12 and 2012/13 snow season.

**5. Data availability**

5

The database presented and described in this article is available for download at Zenodo (Revuelto et al. 2017; https://doi.org/10.5281/zenodo.848277https://doi.org/10.5281/zenodo.579979).

Meteorological data of the AWS are given ready-in .csv format. The meteorological dataset includes observations atim 10-minute-min intervals. . For an easier transferability

- 10 and also to allow a wider post processing, tThe TLS survey point clouds containing the snow depth distribution are available on-line (one file for each TLS acquisition). These files are in ASCII format in the UTM 30T North coordinate system. These point clouds are in the UTM 30T North coordinate system. It is also provided the DEM of the study area in same coordinate system. Cloud-free day photographs from the time-lapse camera
- 15 are available in the online repository, with the correspondence of pixel-ground control points to GPS coordinates. Information on the optics and chip size of the camera is also provided. Additionally all available Melt Out Date distribution maps (MOD, last Julian day with snow presence on each pixel) are included in the database.

20

**6. Summary**

The Izas Experimental Catchment is a well-established study area onin the south face of the Pyrenees, in which different meteorological and snow variables are automatically acquired. Additionally, an importantgreat effort has been made on field data acquisition with TLS-has been conducted during over the last five snow seasons and is ongoing.still maintained. The dataset described here is novel in the Pyrenees because, it represents for the first time, it represents high spatial resolution information on the snowpack distribution and its evolution inon time, as well as making being also available continuous information available on meteorological variables. The high quality of the information obtained has already been-already exploited for different studies on the understanding of snowpack dynamics and <del>on</del>-the improvement of simulation approaches toof snowpack evolution in mountain areas (López-Moreno et al., 2012, 2014, Revuelto et al., 2014b, 2016a, 2016b). However, there exist many scientific questions still go

25

30

unanswered, such as the long term influence of topography on snow dynamics, the

5

spatial distribution of snow during precipitation and strong wind blowing events. Also, the high inter-annual variability of snow accumulation in the Pyrenees has serious important consequences for water management, especially in the Mediterranean area (García-Ruiz et al., 2011). Thus, it is quitevery important to continue obtaining information on snowpack evolution and on-the meteorological variables that controlling

snow dynamics. This information will allow to-the scientific community to better understand processes involved and allowing amake for better adaptation to climate change scenarios. Moreover, offering the possibility of exploiting the information to other fieldscolleges provides, as INARCH does, the opportunity of establishing new

collaboration networks to push forward the frontiers of science limits in mountain areas.

10

**7. Acknowledgments:**

This study was funded by the research projects CGL2014-52599-P "Estudio del manto de nieve en la montaña española y su respuesta a la variabilidad y cambio climatico"

15 (Ministry of Economy and Development, MINECO) and CLIMPY: "Characterization of the evolution of climate and provision of information for adaptation in the Pyrenees" (FEDER-POCTEFA). The authors give thank fors thise unique opportunity forof sharing information throught the International Network for Alpine Research Catchhement Hydrology (INARCH)-. J. Revuelto is supported by a post-doctoral 20 Fellowship fromof the AXA research found 2016 call-(le Post-Doctorant Jesús Revuelto

est bénéficiaire d'une bourse postdoctorale du Fonds AXA pour la Recherche). C. Azorin-Molina is supported by the Marie Skłodowska-Curie Individual Fellowship (STILLING project - 703733) funded by the European Commission

25

 RIEGL Laser Measurements, LPM-321
 Long Range Laser Profil Measurement System

 ©, (2010), Horn, Austria
 ©, (2010), Horn, Austria
 [Contemported]
 [Contemported]

Vicente-Serrano, S.M. (2005). Las sequías climáticas en el valle medio del Ebro:

15 Factores atmosféricos, evolución temporal y variabilidad espacial (Consejo de Protección de la Naturaleza de Aragón).

Vicente-Serrano, S.M. (2006). Spatial and temporal analysis of droughts in the Iberian Peninsula (1910–2000). Hydrol. Sci. J. 51, 83–97.

Vaisala Company, HMP155 Humidity and Temperature Probe specifications © (2012),

20 Ref. B21072EN-E

Viviroli, D., Dürr, H.H., Messerli, B., Meybeck, M., and Weingartner, R. (2007). Mountains of the world, water towers for humanity: Typology, mapping, and global significance. Water Resour. Res. 43, W07447.

Wipf, S., Stoeckli, V., and Bebi, P. (2009). Winter climate change in alpine tundra:

plant responses to changes in snow depth and snowmelt timing. Clim. Change 94, 105–
121.

Young Company, Model 05103-45-5, Wind Monitor- Alpine Model specifications © (2010). R.M. Young Company, Traverse City, Michigan, USA.